

# When disturbances favour species adapted to stressful soils: grazing may benefit soil specialists in gypsum plant communities

Andreu Cera[1,2], Gabriel Montserrat-Martí[3], Arantzazu L. Luzuriaga[4], Yolanda Pueyo[3] and Sara Palacio[1]

[1] Departamento Biodiversidad y Restauración/Instituto Pirenaico de Ecología, Consejo Superior de Investigaciones Científicas, Jaca, Huesca, Spain
[2] Departament de Biologia Evolutiva, Ecologia i Ciències Ambientals, Universitat de Barcelona, Barcelona, Barcelona, Spain
[3] Departamento de Biodiversidad y Restauración/Instituto Pirenaico de Ecologia, Consejo Superior de Investigaciones Científicas, Zaragoza, Zaragoza, Spain
[4] Departamento de Biología y Geología, Física y Química inorgánica, Universidad Rey Juan Carlos, Mostoles, Madrid, Spain

Corresponding authors
Andreu Cera, andreucera@ipe.csic.es
Sara Palacio, s.palacio@ipe.csic.es

## ABSTRACT

**Background:** Herbivory and extreme soils are drivers of plant evolution. Adaptation to extreme soils often implies substrate-specific traits, and resistance to herbivory involves tolerance or avoidance mechanisms. However, little research has been done on the effect of grazing on plant communities rich in edaphic endemics growing on extreme soils. A widespread study case is gypsum drylands, where livestock grazing often prevails. Despite their limiting conditions, gypsum soils host a unique and highly specialised flora, identified as a conservation priority.
**Methods:** We evaluated the effect of different grazing intensities on the assembly of perennial plant communities growing on gypsum soils. We considered the contribution of species gypsum affinity and key functional traits of species such as traits related to gypsum specialisation (leaf S accumulation) or traits related to plant tolerance to herbivory such as leaf C and N concentrations. The effect of grazing intensity on plant community indices (*i.e.*, richness, diversity, community weighted-means (CWM) and functional diversity (FD) indices for each trait) were modelled using Generalised Linear Mixed Models (GLMM). We analysed the relative contribution of interspecific trait variation and intraspecific trait variation (ITV) in shifts of community index values.
**Results:** Livestock grazing may benefit gypsum plant specialists during community assembly, as species with high gypsum affinity, and high leaf S contents, were more likely to assemble in the most grazed plots. Grazing also promoted species with traits related to herbivory tolerance, as species with a rapid-growth strategy (high leaf N, low leaf C) were promoted under high grazing conditions. Species that ultimately formed gypsum plant communities had sufficient functional variability among individuals to cope with different grazing intensities, as intraspecific variability was the main component of species assembly for CWM values.
**Conclusions:** The positive effects of grazing on plant communities in gypsum soils indicate that livestock may be a key tool for the conservation of these edaphic endemics.

## INTRODUCTION

Herbivory by domestic and wild ungulates is one of the main drivers of global vegetation dynamics. Grazing mammals affect plant performance by biomass removal (*Huntly, 1991*) and, accordingly, plants have developed a wide array of adaptations to cope with grazing disturbance throughout evolution (*Díaz et al., 2007*). Grazing is usually considered a crucial biotic filter by restricting the range of trait values to those of species that survive and establish successfully (*Violle et al., 2007*). Grazing can also exert contrasting effects on plant community properties. In productive environments, grazing may alleviate plant-plant competition through changes in competitive hierarchies (*Noy-Meir, Gutman & Kaplan, 1989*; *Louda, Keeler & Holt, 1990*), biomass removal and trampling associated with grazing can also create spatial heterogeneity (*Moret-Fernández et al., 2011*), and thus allow species coexistence due to niche differentiation (*Rosemond, Mulholland & Elwood, 1993*). However, the selective removal of less grazing-tolerant species may also result in a reduction in species richness and diversity (*Milchunas, Sala & Lauenroth, 1988*), particularly when soil resources are scarce and plant productivity is low (*Cingolani, Noy-Meir & Díaz, 2005*).

Extreme soils, such as saline, limestone, serpentine or gypsum, have particular physical and chemical characteristics that restrict plant growth and species distribution (*Rorison, 1960*; *Kazakou et al., 2008*; *Munns & Tester, 2008*; *Moore et al., 2014*). Due to these constraints, atypical substrates are also major drivers of plant evolution (*Hulshof & Spasojevic, 2020*), leading to the development of specialised floras with numerous edaphic-endemics (*Braun-Blanquet, 1932*). Plants have developed different mechanisms to cope with the harsh conditions of extreme soils, and edaphic-endemics are usually soil specialists with substrate-specific strategies (*Kruckeberg & Rabinowitz, 1985*). These strategies allow them to optimise their performance and growth over other plants in their singular habitat (*Cody, 1978*), but may render them less competitive on non-extreme soils, which would explain why edaphic-endemics are frequent in plant communities associated with extreme soils (*Rajakaruna, 2004*).

The assemblage of plant communities on extreme soils has traditionally been explained in relation to plant adaptation to the limiting conditions of these special substrates (*Caçador, Tibério & Cabral, 2007*; *Kazakou et al., 2008*; *Luzuriaga, González & Escudero, 2015*; *Luzuriaga et al., 2020*). Plant communities associated with extreme soils are open shrublands or grasslands (*Brady, Kruckeberg & Bradshaw, 2005*; *Mota et al., 2017*), generally associated with large mammal herbivory and livestock grazing (*Asner & Levick, 2012*; *Bakker et al., 2016*). Consequently, the effect of grazing on plant community composition could be combined with soil restrictions in these systems. Evidence of the effect of grazing on vegetation in extreme soils is controversial. *Ballesteros et al. (2013)* and *Pueyo et al. (2008)* reported negative consequences on the abundance of two gypsum endemic species due to grazing, while other studies found that livestock grazing favoured

edaphic-endemics over other plant species in extreme soils (*Bonis et al., 2005*; *Beck et al., 2015*). Most of these studies analysed the effect of grazing on individual species growing on extreme soils, but more studies at the community level are lacking.

Gypsum plant communities offer an excellent study system to evaluate the joint effect of extreme soils and grazing on plant community assemblage. The weathering of gypsum rock generates an unusual soil with high Ca and S content and low P availability that severely limits plant life due to nutrient imbalances (*FAO, 1990*; *Casby-Horton, Herrero & Rolong, 2015*). Gypsum soils occur worldwide in drylands, where extensive grazing almost always occurs (*Pueyo et al., 2008*; *Akhani, 2015*). The flora associated with gypsum is a unique endemic flora identified as an international conservation priority (*Moore et al., 2014*; *Escudero et al., 2015*), composed by edaphic-endemics, which are soil specialists (*Palacio et al., 2007*), and also species with wide ecological ranges (*Meyer, 1980*). Gypsum endemics show generally high affinity for gypsum soils (*Luzuriaga, González & Escudero, 2015*), also referred as gypsophily value (*Mota, Sánchez-Gómez & Guirado, 2011*; *Musarella et al., 2018*), and have foliar chemical composition characterized by high foliar S-accumulation (*Duvigneaud & Denaeyer-De Smet, 1966*; *Salmerón-Sánchez et al., 2014*; *Merlo et al., 2019*). *Braun-Blanquet & de Bolòs (1957)* suggested that plant communities rich in edaphic-endemics might be favoured by moderate grazing in gypsum soils, as grazing hardens soil conditions and impedes the weathering of gypsum, preventing the formation of a more organic soil that would favour other plant communities without gypsum endemics. However, no previous studies have evaluated the effect of livestock grazing on the assembly of species with different affinity for gypsum soils, or its relationship to the foliar composition of gypsum plants.

In this context, functional traits may play a key role in species composition on extreme soils with different intensities of herbivory. Plant functional traits that favour persistence under grazed conditions can be classified into avoidance and tolerance mechanisms (*Briske & Richards, 1995*). Avoidance mechanisms include traits that reduce plant accessibility and palatability, whereas tolerance traits lead to increased growth rate to compensate for biomass loss due to grazing. Leaf C and N content indicate species tolerance to grazing (*Capó et al., 2021*), where high N content is generally related to high growth rates (*Pérez-Harguindeguy et al., 2013*) and thus, to species that are able to compensate the biomass lost by herbivory (*Grime, 2006*). Moreover, leaf S content is a functional trait related to plant specialisation to gypsum soils (*Merlo et al., 2019*; *Cera et al., 2021*), although its ecological significance remains unknown (*Palacio et al., 2007*). High leaf S content could play a significant role in grazing-avoidance in gypsum environments (*Palacio et al., 2014*), because foliar S accumulation has been related to herbivore-deterrent compounds in Brassicales as glucosinlates (*Ernst, 1990*), and in some species of *Acacia* as crystals with S (*He et al., 2015*).

The aim of this study was to evaluate the extent to which the assembly of perennial plant communities under different grazing intensities and on high gypsum soils, is mediated by the affinity of species for gypsum, by species traits related to gypsum adaptation (leaf S concentration) or by traits related to plant tolerance to herbivory such as leaf C and N concentrations. If stressful conditions derived from atypical gypsum soils are the main

selective force shaping plant communities on gypsum, we would expect gypsum-endemics to dominate over other species, independently of the grazing pressure. In this sense, the assembly of plant communities in gypsum environments would mainly depend on the gypsum affinity of plants (*Luzuriaga, González & Escudero, 2015*, *Luzuriaga et al., 2020*) and species with high S leaf content would be always dominant in plant assemblages. However, if grazing is the main factor driving plant assembly on gypsum plant communities, species with a rapid-growth strategy (*i.e.*, species with high leaf N content and tolerance mechanisms, *Grime, 2006*) and/or with deterrent traits (avoidance mechanisms, *Briske & Richards, 1995*) would be dominant under high grazing pressure in relation to non-grazed conditions. This would lead to increased values of community weighted mean (CWM) for leaf N and decreased leaf N functional diversity in the most grazed plant communities. Finally, if as expected, herbivores have historically played an important role in shaping plant assembly in extreme soils of the Mediterranean region (*Braun-Blanquet & de Bolòs, 1957*; *Montserrat-Martí & Gómez-García, 2019*), edaphic-endemics should have developed both mechanisms to tolerate or avoid grazing and mechanisms to persist in restrictive soils. In this context, species with high affinity for gypsum and traits to cope with restrictions typical of gypsum soils (*i.e.*, leaf S-accumulation), also fitted with traits to tolerate (*i.e.*, high growth rates) or avoid herbivory would be favoured under grazing conditions, increasing CWMs values of leaf S and gypsophily value (GV) and decreasing their functional diversity in the most grazed conditions.

We also evaluated the contribution of intraspecific trait variability to cope with different levels of herbivory during the species assembly process. For this purpose, we applied the method proposed by *Lepš et al. (2011)* and *de Bello et al. (2011)* to disentangle the effects of interspecific *vs* intraspecific trait variability on species assembly. In this context, if species adapted to gypsum are also adapted to grazing disturbance, we would expect responses at the population level (*i.e.*, intraspecific variability), since species would have enough intraspecific variability to cope with changes in grazing intensity. Otherwise, if species adapted to gypsum are not necessarily adapted to cope with grazing disturbance, we would expect those species that cannot withstand grazing would disappear when grazing intensity increases. This would lead to a shift in species composition (inter-specific variability) depending on their ability to cope with herbivory regardless of their soil affinity, so we would expect species assembly to be mainly due to species turnover.

## MATERIALS AND METHODS

### Study site

This study was conducted in three locations in the Middle Ebro Valley (NE Spain): Pedriza (Mediana de Aragón 41°24′17″N, 0°41′20″W), Corral del Hoyo (Mediana de Aragón 41°25′38″N, 0°44′45″W) and Valdemolino (Mediana de Aragón 41°27′30″N, 0°45′31″W) (Figure S1). All of them have gypsum soils as their main lithology and have a semi-arid Mediterranean climate with mean annual temperature of 14.9 °C and mean total annual rainfall of 353.9 mm yr-1 (data from the nearest weather station at Farlete 41°50′56″ N, 0°30′19″ O). The landscape in this area consists mainly of low hills (480 m.a.s.l. average)
and flat-bottomed valleys, which are currently cultivated (*Foronda et al., 2019*). Above-ground vegetation in the three locations was predominantly composed of shrubs, forbs and grasses, like *Brachypodium retusum* (Pers.) P.Beauv., *Gypsophila struthium subsp. hispanica* (Willk.) G.López, *Helianthemum squamatum* (L.) Dum.Cours., *Herniaria fruticosa* L. and *Plantago albicans* L. The vegetation structure in our study sites was a matrix of plant patches and bare soil, with total vegetation cover of 25.45% ± 12.97 on average. The gypsum outcrops of the Middle Ebro Valley have a large legacy of extensive grazing practices, although these have now been drastically reduced in most areas (*Braun-Blanquet & de Bolòs, 1957*; *Pueyo et al., 2008*).

## Plant community surveys

We conducted vegetation surveys in June 2018, a year with wetter than average conditions according to a multiscale drought index (SPEI, (*Vicente-Serrano et al., 2017*)). We carried out three independent grazing gradients, one in each location, to reduce the influence of other environmental variables. In addition, locations were selected in the same region to avoid climatic biases. Gradients were established by selecting three flat hilltop sites with different grazing intensity in each location (Figure S1). Each gradient included one site with no grazing for a few decades (hereafter referred to as low grazing), one site with medium grazing and one site with high grazing pressure near the pens. The current grazing intensity was estimated by interviewing local farmer, as described in *Pueyo et al. (2008)*, with whom ongoing communication is maintained. Thirty-five 2 × 2 m plots were randomly established within each grazing intensity site (5,000 $m^2$) in each of the three locations (35 plots × 3 grazing intensities × 3 locations; $N = 315$ plots). In each plot, every species occurrence was recorded and species cover visually estimated. All plant taxa present on plots were listed, and a score was derived based on species cover in that plot. Cover, as defined for our purpose, is the fraction of the total plot area that is occupied by a particular species when viewed from directly above. Species were identified in the field and revised taxonomically in the laboratory using specific literature (*Castroviejo S (coord. gen.), 1986*; *Aizpuru et al., 1999*). Nomenclature followed The International Plant Names Index (*IPNI, 2021*). In addition, total vegetation cover and maximum vegetation height were measured per plot.

## Soil sample collection and analyses

Three soil samples per site were collected from 5 to 15 cm depth, removing the surface crust ($N = 27$), to characterise soil physicochemical properties (Table S1). All soil samples were air dried for 2 months and subsequently sieved through a 2 mm sieve before physical and chemical analyses. Gypsum content was measured according to (*Artieda, Herrero & Drohan, 2006*). Soil texture was determined with a particle laser analyser (Mastersizer 2000 Hydro G, Malvern, UK). Soil pH and conductivity were measured with a pH/conductivity meter (Orio StarA215, Thermo Scientific, Waltham-MA, USA) by diluting samples with distilled water to 1:2.5 (w/v) and 1:5 (w/v), respectively. A subsample of each sieved soil was finely ground using a ball mill (Retsch MM200, Restch GmbH, Haan, Germany) and subsequently used to analyse the elemental concentrations of N and C with an elemental

**Table 1 Total cover per plot.** Total cover (%) and standard error per plot of all the species with functional traits values.

| Locality | Low | Medium | High |
|---|---|---|---|
| Corral del Hoyo | 59.9 ± 4.7 | 82.4 ± 5.2 | 100.0 ± 3.2 |
| Pedriza | 67.2 ± 2.4 | 67.2 ± 4.4 | 56.5 ± 4.2 |
| Valdemolino | 93.9 ± 3.5 | 84.2 ± 4.1 | 80.2 ± 2.4 |
| Averaged total | 73.7 ± 3.5 | 77.9 ± 2.7 | 80.1 ± 2.7 |

analyzer (TruSpec CN, LECO, St. Joseph-MI, USA), elemental analyses were performed by EEZ-CSIC Analytical Services.

## Gypsophily value

All perennial species recorded were classified by their affinity to gypsum soils using the gypsophily value (hereinafter GV; *Mota, Sánchez-Gómez & Guirado, 2011*), as performed in *Luzuriaga et al. (2020)*. The GV was calculated by a group of experts on gypsum flora of the Iberian Peninsula, using the Delphi technique (*Mota et al., 2009*). The GV ranges from 1 (species that avoid gypsum soils), 2 (species that have no preference for gypsum soils, but grow on them), and from 3 to 5 for species with a preference for gypsum soils, with 5 for strict gypsum species.

## Measures of plant traits

Leaf traits were measured in each level of grazing (*i.e.*, low, medium, high) in one location (Valdemolino) which is the unique location with at least five replicates per species at each site, with healthy adult plants. We measured on 14 perennial species, which were present in the three sites with different grazing intensities in the location and accounted for 75.6% of total plant cover (Table 1), although they were not present in all plots. The leaf traits measured in Valdemolino were used to calculate community indices in the three locations (explained below), using them as habitat-specific traits of each intensity of grazing. Leaf samples were collected from five different individuals per species in each site. Mature, non-senescent and undamaged leaves were collected. To assess the N, C and S concentrations in leaves, leaf samples were dried to a constant weight at 50 °C during 5 days and subsequently finely ground using a ball mill (Retsch MM200, Restch GmbH, Haan, Germany). N, C and S were analysed with an elemental analyser. N concentrations were analysed in EEZ-CSIC Analytical Services (TruSpec CN, LECO, St. Joseph-MI, USA), and C and S concentrations were measured in IPE-CSIC Analytical Services (TruSpec CNS, LECO, St. Joseph-MI, USA).

## Community level indices and statistical analyses

All statistical analyses and graphics were performed using R version 4.0.2. To characterise gypsum affinity and leaf trait values (S, N and C leaf contents) at the community level, we calculated the gypsophily value in each plot for gypsum affinity and Community Weighted Means (CWM), and Functional Diversity indices (FD) for leaf traits. The CWM quantifies

the mean contribution of species with different traits to each species assemblage (*Garnier, Navas & Grigulis, 2016*). It was calculated as:

$$CWM = \sum_{i=1}^{n} p_i trait_i \tag{1}$$

where pi represents the plant cover of species i and traiti the value of that specific trait for the species i. We used the *dbFD* function in the FD package version 1.0-12 (*Laliberté & Shipley, 2011*).

The FD evaluates the dispersion of trait values in the community. It was calculated as:

$$FD = \sum_{i=1}^{n} \sum_{j=1}^{n} p_i p_j d_{ij} \tag{2}$$

where pi and pj represent the plant cover of species i and j, respectively; dij the distance between both species in the functional space. We used the quadratic diversity of RaoQ (*Botta-Dukát, 2005*) using the *melodic* function (*de Bello et al., 2016*) and Gower dissimilarity matrices of species-specific trait values. Also, we calculated Simpson index using the *melodic* function, as:

$$E = 1 - \sum_{i=1}^{n} p_i^2 \tag{3}$$

A PERMANOVA based on Bray-Curtis distances and type III Sum of Squares was performed to assess differences in species composition among the 315 plots using *adonis* function in the vegan package version 2.4-6 (*Oksanen et al., 2007*). To prepare data for PERMANOVA, species that appeared in less than 5% of the plots were removed and cover data were square root-transformed, to avoid statistical biases of rare species. Grazing intensity was considered as a fixed factor with three levels, and location as strata. Soil properties were included in the model as a covariate. This covariate was the main ordination axis of a Principal Component Analysis (PCA) performed with the soil physicochemical features measured at each site after scaling all variables using the *rda* function in the vegan package version 2.4-6 (*Oksanen et al., 2007*). Non-metric Multidimensional Scaling (NMDS) was used to represent relationships among species composition, environmental features (pH, conductivity, soil C, soil N, gypsum content, and sand, loam and clay proportions), and grazing intensity of plots. We used cover data of the 30 species from the 315 plots and *metaMDS* and *envfit* functions in the vegan package version 2.4-6 (*Oksanen et al., 2007*).

The effect of grazing intensity on plant community properties was evaluated using Generalised Linear Mixed Models (GLMM) with grazing intensity as a fixed factor, and plot nested into location as random factors. We modelled eleven response variables at the community level: total vegetation cover, maximum canopy height, taxonomic diversity (Simpson index), gypsum affinity and CWM and FD indices for three leaf functional traits (S, N and C contents) (see below for CWM for gypsum affinity species). Over- and under-dispersion and normality of models residuals were detected using *simulateResiduals*

function in the DHARMa package version 0.3.1 (*Hartig, 2017*). *lmer* function was applied when normality of residuals was fulfilled, otherwise the Gamma distribution was applied using the *glmer* function in the lme4 package version 1.1.5 (*Bates et al., 2015*) or, in case residuals were over- or under-dispersed, we used the *glmmTMB* function in the glmmTMB package version 1.1 (*Magnusson et al., 2019*). F-test was used to test the significance of GLMM, except when using the *glmmTMB* function. When differences were statistically significant, we assessed multiple comparisons among levels of grazing intensity with the *glht* function in the multcomp package version 1.4-13 in R (*Hothorn, Bretz & Hothorn, 2009*), applying a Bonferroni correction. In the case of CWM for gypsum affinity of species, the standard regression analysis on species-specific traits could increase type I error rates in the CWM approach (*Braak, Peres-Neto & Dray, 2018*). Thus, we used the max test approach (row- and column-based permutation) to explore the correlation of CWM with grazing intensity (*Zelený, 2018a*), using *test_cwm* function in the weimea package version 0.1.4 (*Zelený, 2018b*). On the other hand, we used a standard parametric test for leaf content CWM values, because it is not clear that habitat-specific CWM values suffer from the same bias, as no species-level value is applied to all plots (*Zelený, 2018a*).

We used the method proposed by *Lepš et al. (2011)* and *de Bello et al. (2011)* to disentangle the effects of interspecific *vs* intraspecific trait variability on species assembly processes. We calculated three components for each index (CWM and FD) per plot: (1) The "Fixed" component: was calculated using the mean value of each trait measured over all 15 individuals of that species in the three grazing intensity levels of the study; (2) The "Specific" component: was calculated using the trait average values of each species measured over all five individuals of that species at that particular level of grazing intensity; (3) The "Intra-specific" component: was calculated as the average difference between specific and fixed values. Sum of Squares was calculated for each component and trait using a parametric method (lmer function) with plot nested within location as random factors when residuals fitted a normal distribution, and a non-parametric one (*adonis* function with Euclidean distance) with location as strata, otherwise. Finally, we plotted the Sum of Squares decomposition of each CWM and FD values on grazing intensity to understand the relative contribution of interspecific variation, intraspecific trait variation and their covariation in community composition under different grazing intensities.

## RESULTS

### Variation in species composition

We found 52 perennial plant species (Table S2). Total plant cover and canopy height decreased from low to high grazing intensities (Fig. 1, Table 2). Plant composition was significantly different among grazing levels (*F-ratio* = 17.213, *P-value* = 0.001, *TVE* = 9.68%), as well as in response to soil features (*F-ratio* = 10.37, *P-value* = 0.001, *TVE* = 2.92%; Figure S2, Table S3). Specifically, Soil C ($R^2$ = 0.41) and gypsum content ($R^2$ = 0.32) explained the largest proportion of variability in plant species composition (Table S3).

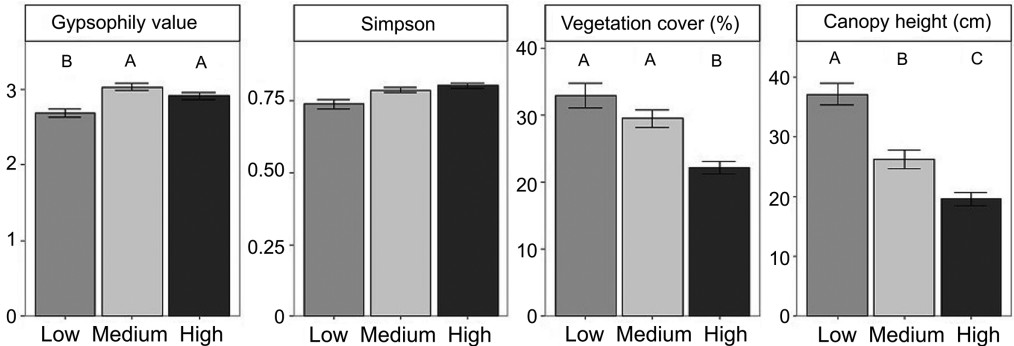

**Figure 1 Differences in community properties and the gypsophily value among grazing intensity levels.** Differences in community properties and the gypsophily value among grazing intensity levels. Mean values ± standard errors are represented. Different letters indicate significant differences among levels of grazing intensity after multiple comparison tests with Bonferroni correction (*P* < 0.05).

**Table 2 Effect of grazing intensity on the main features of plant community properties and the gypsophily value after GLMMs.** Effect of grazing intensity on the main features of plant community properties and the gypsophily value after GLMMs. Chi-square values obtained by Wald tests based on generalised linear mixed models plus the family of error distributions and link functions assumed in the models are indicated. Id: identity link function; Log: logarithmic link function. Superscripts indicate GLMMs were run with the glmmTMB function to correct for dispersion of residuals.

|  | Family (link) | Df | Chis-square | Pr(>Chisq) |
|---|---|---|---|---|
| Vegetation cover (%) | Gamma (Log)[1] | 2 | 41.889 | 0.001 |
| Canopy height (cm) | Gamma (Id)[1] | 2 | 76.541 | 0.001 |
| Simpson Index | Binomial[1] | 2 | 1.506 | 0.471 |
| Gypsophily value | Gaussian | 2 | 30.32 | 0.001 |

## Gypsum affinity and leaf traits (C, N, S-contents) at the community level

Gypsum specialists were favoured over non-specialists with increasing grazing (Fig. 2, see means and SE between locations in Figure S3). Our results showed higher CWM of gypsophily values (referred to GV) on the most heavily grazed plots than on the less grazed ones using a standard test (Table 3). However, these results may be slightly optimistic: when we applied a correction to avoid type I error (*i.e.*, max test) the row-based test was statistically significant (*P-value* = 0.001), but the column-based test was not (*P-value* = 0.413). Plants with a rapid-growth strategy (high leaf N, low leaf C) and high leaf S were more likely to assemble in the most grazed plots, since CWM values of leaf C decreased and CWM values of leaf N and leaf S increased in higher grazing intensities (Fig. 2, Table 3). Medium grazing intensity was the most heavily filtered level on leaf C and leaf N, since plant communities showed the lowest FD values. Contrastingly, FD values of leaf S were statistically different (Table 3) and decreased with increasing grazing pressure (Fig. 2, *P-adjusted* = 0.06 between High and Low intensity). However, all plots along the

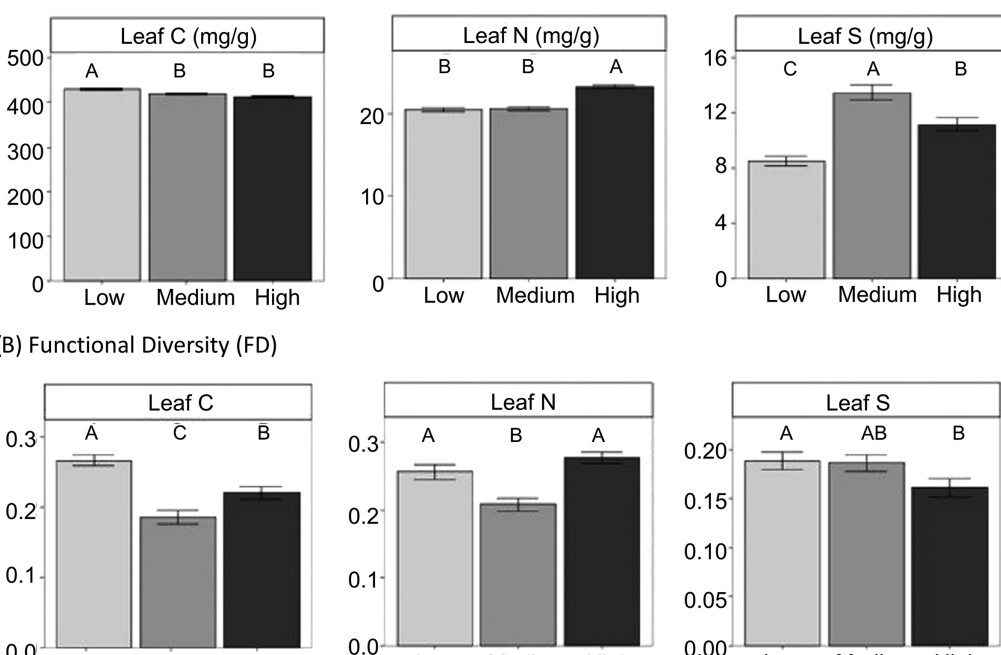

(A) Community Weighted-Means (CWM)

(B) Functional Diversity (FD)

**Figure 2 CWM and FD of leaf traits at the community level in different grazing intensities.** CWM and FD of leaf traits at the community level in different grazing intensities. Means ± standard errors are shown. Different letters indicate significant differences among levels of grazing intensity (low, medium and high) after multiple comparisons with Bonferroni correction ($P < 0.05$).

**Table 3 Results of generalised linear mixed models (GLMMs) for community weighted mean (CWM) and functional diversity indices.** Results of generalised linear mixed models (GLMMs) for community weighted mean (CWM) and functional diversity indices. Chi-square values obtained by Wald test based on GLMMs. The family of error distributions and link functions assumed in each model are also indicated. Id: identity link function; Log: logarithmic link function. FD: functional diversity values. Superscript indicate analysed with glmmTMB function to correct for dispersion of residuals.

| | *Community-weighted mean* | | | |
|---|---|---|---|---|
| | **Family (link)** | **Df** | **F-ratio** | **Pr(>F)** |
| Leaf C | Gaussian | 2 | 18.23 | 0.001 |
| Leaf N | Gaussian | 2 | 67.34 | 0.001 |
| Leaf S | Gaussian | 2 | 31.81 | 0.001 |
| | *Functional diversity* | | | |
| | Family (link) | Df | F-ratio | Pr(>F) |
| Leaf C | Gaussian | 2 | 22.06 | 0.001 |
| Leaf N | Gaussian | 2 | 31.995 | 0.001 |
| Leaf S | Gamma (Id) | 2 | 6.7121 | 0.035 |

gradient were heavily filtered, since FD values of leaf traits were generally lower than the Simpson Index (leaf C = 0.22 ± 0.01, leaf N = 0.25 ± 0.01, leaf S = 0.18 ± 0.01), which was equal along the gradient (Mean = 0.78 ± 0.01).

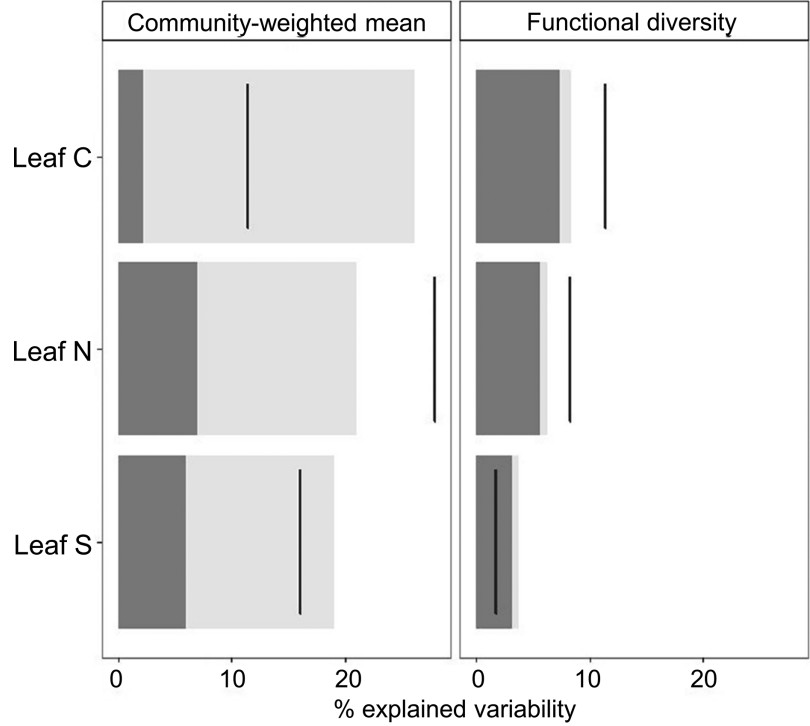

**Figure 3 Decomposition of the variability in CWM and FD values explained by grazing intensity following** *Lepš et al. (2011)*. Decomposition of the variability in CWM and FD values explained by grazing intensity following *Lepš et al. (2011)*. The dark grey portion of bars corresponds to the contribution of interspecific variability and the light grey portion to intraspecific effects. Black lines denote total variation. The ranges between the top of the bar and the black line correspond to the effect of covariation between inter and ITV; if the line is right to the bar the covariation is positive, and if the line crosses the bar, the covariation is negative.

Overall, the process of species assembly in different grazing intensities was mainly determined by intraspecific shifts in functional traits (Fig. 3), since CWMs of leaf traits along the grazing intensity gradient were largely explained by ITV. Contrastingly, the amplitude of the range of trait values that occur in different grazing intensities was explained by species turnover, since shifts in FDs of studied leaf traits were explained mainly by interspecific trait variability (Fig. 3). Further, the contribution of interspecific and ITV on CWM and FD values were positively correlated for leaf N and negatively for leaf S. Leaf C showed a positive correlation for FD and negative for CWM values (Fig. 3).

## DISCUSSION

Our results provide evidence that livestock grazing may benefit soil specialist species during community assembly. Specifically, the relative abundance of gypsum specialists and of the leaf S content of species increased in species assemblages under medium and high grazing intensities. Although some studies found that certain soil specialists are more vulnerable to herbivory than their non-specialist relatives (*Dechamps et al., 2008*; *Kay et al., 2011*; *Strauss & Boyd, 2011*), our results highlight that under medium grazing pressure species with higher affinity for gypsum soils were favoured. Our results showed a

clear trend of increasing relative abundance of gypsum specialists along the grazing gradient, although some caution should be applied to the statistical interpretation of this result. The sampling design should include more replicates of grazing level per site rather than using pseudoreplicates (*i.e.*, plots), which could result in low β-diversity within sites, and increase type I error rates in the CWM approach (*Zelený, 2018a*). Further, the trend of increasing relative abundance of gypsum specialists was different between the three locations studied, although the highest values of GV were always reported in the medium and high grazing intensity site. Despite these concerns, our study is a remarkable result that aligns with previous studies of plant communities growing on extreme soils, such as white sands of tropical forests (*Fine, Mesones & Coley, 2004*), serpentine grasslands (*Beck et al., 2015*) and saline soils (*Bonis et al., 2005*). All these studies found that the occurrence of edaphic specialists was dependent on herbivory, most likely because herbivores modified competitive plant-plant interactions (*Louda, Keeler & Holt, 1990*; *Grover & Holt, 1998*) and may benefit the less competitive soil specialist species.

Our study showed that species prone to accumulate S in their leaves were favoured in medium and highly grazed conditions (*i.e.*, larger CWM values for the leaf-S trait). Leaf S-accumulation is usually related to gypsum specialist species (*Merlo et al., 2019*), but the specific ecological role of S-accumulation in gypsum plants remains unknown (*Palacio et al., 2007*). It has been described that a high translocation of the excess element in soil to plant tissues (*i.e.*, S in gypsum soils) could be a strategy to optimise plant growth in extreme soils by avoiding interference of that specific element with plant metabolism (*Kabata-Pendias, 2010*; *Tran et al., 2020*). Other studies proposed that high leaf S content could be related to a herbivore-deterrent strategy to avoid biomass loss in nutrient-limited habitats (*Ernst, 1990*; *Palacio et al., 2014*), as proposed for some species of Brassicales and *Acacia* (*He et al., 2014*; *Tuominem et al., 2019*). These results point at a selection of increased foliar S accumulation in plants growing on gypsum as a mechanism to deter herbivores (*Boyd, 2007*; *Hoerger, Fones & Preston, 2013*). These results are compatible with a potential role of herbivory as a selective force underlying the evolution of edaphic specialists (*Fine et al., 2006*; *Lau et al., 2008*), promoting the selection of foliar S accumulation in gypsum specialists. Therefore, it could be interpreted that foliar S accumulation could be an adaptive strategy of gypsum specialists to obtain better performance in gypsum environments disturbed by herbivores, where nutritional and water stresses are also present.

Some authors suggested that when resource availability is scarce, the costs of losing plant tissues due to herbivory are high, and plants that invest in chemical defences should be selected (*Coley, Bryant & Chapin, 1985*), mainly because the ability to compensate biomass losses is dependent on resource availability (*Strauss et al., 1999*). Thus, tolerance to herbivory is expected to be low in edaphically stressful substrates. Nevertheless, our results showed that species with higher leaf-N contents were selected in highly grazed compared to low grazed conditions (*i.e.*, greater CWM of leaf-N and low leaf-C in highly grazed sites), suggesting that species with comparatively higher growth rates under the low resource availability of gypsum soils may have been favoured under grazing. These species could also be soil specialists. In the case of gypsum environments, soil specialists are

expected to perform better in their atypical substrate than other soils (*Cera et al., 2021*). Also, they are more likely to assemble in gypsum soils than other soils (*Luzuriaga, González & Escudero, 2015*), as we observed high relative abundance of gypsum specialists along the gradient, regardless of grazing pressure. High foliar N concentrations could also be related to the accumulation of N-rich unpalatable compounds, more aligned with a grazing-avoidance strategy (*Tuominem et al., 2019*). However, such compounds are also frequently C-rich (*Tuominem et al., 2019*), which would have resulted in higher C concentrations, contrary to our observation of decreased foliar C in high grazing plots.

The effect of grazing on gypsum plant communities varied with grazing intensity. Low and medium grazed plots showed similar plant cover, but medium grazed plots displayed lower FD values (narrower range of trait values) than low and high grazed plots for leaf C and leaf N. Both leaf traits are linked to plant growth strategy (*Grime et al., 1997*; *Pérez-Harguindeguy et al., 2016*), indicating a narrow range of growth rate values of the species that assemble in medium grazed plots. These results can be explained by the effect of different grazing intensities on plant competiveness in the stressful conditions of gypsum soils. The higher FD in low grazed plots can be explained by the hypothesis of limiting similarity (*Abrams, 1983*). In these plots, there is high competition for resources due to the nutrient scarcity of gypsum soils (*Boukhris & Lossaint, 1970*). Species with different N requirements can coexist, because N acquisition niches do not overlap (*Montesinos-Navarro et al., 2017*), leading to higher FD values of traits related to growth strategy than medium grazed plots. Contrastingly, the top-down effect of sheep in high grazed plots seems to be due to disturbance associated to grazing that may reduce the biomass of dominant species (*Noy-Meir, Gutman & Kaplan, 1989*), alter the harsh physical crust typical of gypsum soils (*Moret-Fernández et al., 2011*) and eventually create new gaps for colonisation (*Rosemond, Mulholland & Elwood, 1993*). These processes may allow species with contrasting growth rates to coexist in heavily grazed conditions.

The relative contribution of species turnover and intraspecific trait variability on the species assembly process at different grazing intensities (*Lepš et al., 2011*) has been poorly analysed. Our results showed that intraspecific variability was the main component of species assembly for CWM values, while shifts in FD resulted from species turnover. The design of this study did not allow checking whether intraspecific variation was due to plastic responses of plants or heritable differences between genotypes (*Bolnick et al., 2011*). Furthermore, the design of our study limited the exploration of interspecific trait variation, as we analysed leaf traits in only a subset of 14 species growing along the gradient. The high relevance of ITV indicates that the species that finally conformed gypsum plant communities had enough functional variability among individuals to cope with different grazing intensities. Considering that livestock has been a usual anthropogenic activity since the Neolithic in the Iberian Peninsula (*Balaguer et al., 2014*), our results are compatible with the hypothesis that grazing has acted as an evolutionary driver over time, promoting a regional species pool fitted with successful strategies and enough functional variability among individuals to cope with herbivory. Consequently, in our study system, grazing did not act strictly as a biotic filter selecting certain species and jeopardizing others, but it acted in a subtler way, modifying the range of values of each plant trait selected during plant

community conformation. This is a remarkable result that aligns with previous studies on the effect of herbivory on plant community assembly in environments with a long evolutionary history of grazing, such as Tibetan alpine meadows (*Niu et al., 2016*), and Inner Mongolia grasslands (*Zheng et al., 2015*).

# CONCLUSIONS

To sum up, our results are compatible with the notion that grazing has likely been a powerful evolutionary driver in the conformation of plant assemblages on gypsum soils. Our results seem to indicate that herbivores may promote plant edaphic specialists in gypsum soils. Extensive livestock grazing should be considered as key tool to promote plant communities in gypsum soils where endemics persist, and should be evaluate with long-term studies, monitoring the intensity of livestock to avoid over-grazing, and linked other factors, such as climatic, that could affect the assemblage of communities.

# ACKNOWLEDGEMENTS

We are grateful to Natalia Revilla for help with vegetation surveys, to Nate Heiden and Antonio Palma for help with plant collection, to Elena Lahoz, José Azorín and María Pérez-Serrano Serrano for help with plant and soil analyses, and to Pablo Tejero for help with map image. Mehdi Abedi and Khadijeh Bahalkeh provided useful comments on the results. Rebecca E. Drenovsky provided valuable comments on earlier versions of this manuscript.

## Funding

This work was supported by Gobierno de España (MICINN, CGL2015-71360-P, CGL2016-80783-R and PID2019-111159GB-C31); by the European Union's Horizon 2020 (H2020-MSCA-RISE-777803); and by Consejo Superior de Investigaciones Científicas (COOPB20231). Andreu Cera and Sara Palacio were funded by an FPI fellowship (MICINN, BES-2016-076455) and a Ramón y Cajal Fellowship (MICINN, RYC-2013-14164), respectively. The publication fee was supported by the CSIC Open Access Publication Support Initiative through its Unit of Information Resources for Research (URICI). The funders had no role in study design, data collection and analysis, decision to publish, or preparation of the manuscript.

## Grant Disclosures

The following grant information was disclosed by the authors:
Gobierno de España: MICINN, CGL2015-71360-P, CGL2016-80783-R and PID2019-111159GB-C31.
European Union's Horizon 2020: H2020-MSCA-RISE-777803.
Consejo Superior de Investigaciones Científicas: COOPB20231.
Gobierno de España Fellowship: MICINN, BES-2016-076455.
Ramón y Cajal Fellowship: MICINN, RYC-2013-14164.

## Competing Interests

The authors declare that they have no competing interests.

## Author Contributions

- Andreu Cera conceived and designed the experiments, performed the experiments, analyzed the data, prepared figures and/or tables, authored or reviewed drafts of the article, and approved the final draft.
- Gabriel Montserrat-Martí conceived and designed the experiments, performed the experiments, authored or reviewed drafts of the article, and approved the final draft.
- Arantzazu L. Luzuriaga conceived and designed the experiments, analyzed the data, authored or reviewed drafts of the article, and approved the final draft.
- Yolanda Pueyo conceived and designed the experiments, authored or reviewed drafts of the article, and approved the final draft.
- Sara Palacio conceived and designed the experiments, performed the experiments, analyzed the data, authored or reviewed drafts of the article, and approved the final draft.

## Data Availability

The raw data and R code that support the findings of this study are available in the Supplemental Files.

## Supplemental Information

Supplemental information for this article can be found online at http://dx.doi.org/10.7717/peerj.14222#supplemental-information.

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
