# Peer review of "When disturbances favour species adapted to stressful soils: grazing may benefit soil specialists in gypsum plant communities"

_PeerJ, doi:10.7717/peerj.14222_

## Round 0.1 · original submission · Major Revisions

Your paper presents interesting and well-presented findings. However, the Reviewers brought up legitimate concerns that first need to be addressed.

Reviewer 1 ·

Basic reporting

In the research paper entitled “When disturbances favour species adapted to stressful soils: grazing may benefit soil specialists in gypsum plant communities”, the authors perform a study about some ecological patterns that could influence the Iberian gypsophile flora. This is a very interesting research work that applies a noteworthy statistical methodology in order to establish multiple relationships between biotic and elemental characteristics of endemic flora species on Spanish gypsum outcrops, and test the effect of the grazing on them. The level of endemicity of the gypsophile flora of the Iberian Peninsula is outstanding, and a number of studies has been published about this. The draft is very interesting, well-structured and written; however, I would like to encourage the authors to address some of the following issues.

Figures. Study area. Would be possible to show a map of the study areas including the location of the sampled plots? The gypsum outcrops in Spain are very well mapped, so it would be very interesting to locate the sampling points on the map of all Spanish gypsum outcrops so that the scope of the fieldwork can be observed. All maps should contain a legend, a scale, an indication of N, and a grid with geographic coordinates for correct location.

Experimental design

Was the estimation of stocking density based on the Pueyo (2013) study? And if so, how can we explain the relationship between livestock density data from 2013 and vegetation cover data from 2018, i.e. has the livestock density not changed in the intervening five-year period? It would be necessary to provide more data on the estimation of stocking density and how it has been related to vegetation sampling. Please, this aspect should be clarified and more data should be provided in the manuscript.

Why were sample plots of 2x2 meters chosen? Among the species selected for the study are Gypsophila struthium or Genista scorpius, which are species capable of reaching medium size and for which this plot size is somewhat small. It is also not understandable that 35 plots x 3 grazing intensities x 2x2m2 =420 m2 would represent a sampling effort capable of covering an area of 5000m2, within each grazing intensity site. Please explain this aspect, as it results unclear.

Please explain why leaf tissues were analyzed from only one location. This may be one of the major methodological difficulties of the research, since if the rest of the data have been obtained for all sites, why have leaf tissues not been analyzed as well? The most coherent approach would seem to be to analyze data from the same 14 species at the three sampled locations and for the different grazing intensities in order to be able to draw conclusions that can be generalized to at least the three sites studied.

Please include information about the reason to collect these 14 flora species to obtain the N, C and S values (Table S4). In this sense, the selection of flora species may be another major drawback in the methodological framework. As for the gypsophily index of the collected species, nine of these 14 plant species have a value of “2”, compared to only four species with gypsophily values higher than “4”, and one species (Sideritis cavanillesii) that does not appear in Table S2. Considering that, and according to the cited reference in the text, a gypsophile species would be considered at a level higher than “3.5” (other authors points to values higher than “4”), we can observe that the analysis associated to the N, C and S contents has been carried out with more than 70% of species not considered as true gypsophiles. This factor can be decisive when interpreting the results by trying to establish relationships for the flora unique (or endemic) on gypsum substrates. In addition, this fact, coupled with that only leaf content data from one sampling location is provided and the level of gypsophily of the selected plants does not fit in general with a true gypsophile, may not support a conclusion about the benefit of soil specialist species during community assembly. However, it could support the evolution of the plant community on gypsum under the conditions studied.

Please revise the scientific names, there are several mistakes in table S4.

Validity of the findings

In view of the data provided and the results obtained, it seems that the conclusions exposed may not be fully supported, or perhaps these conclusions cannot be extrapolated to all presence of flora on these extreme substrates, or be absolutely generalizable, as it is possible that other plant communities that develop on gypsum, or other bizarre substrates, have not evolved with the same level of factors associated with stocking rates as the ecosystems of the Mediterranean Basin, occupied for millennia by modern societies.

Taking into account that the vegetation sampling was only carried out in year 2018, maybe a single year of sampling would not be sufficiently representative to extrapolate conclusions, but it could indicate that in the three cases studied the behavior of the vegetation could be subject to the conditions shown. And that this behavior could be one of several patterns of evolution of gypsum vegetation that can occur in the world.

It could be suggested that more in-depth field work would require studies with permanent plots and monitoring of stocking rates over several consecutive years in which other factors such as climatic factors could also be measured.

Reviewer 2 ·

Basic reporting

The article titled “When disturbances favour species adapted to stressful soils: grazing may benefit soil specialists in gypsum plant communities” is a research article dealing with a very important issue as the adaptation of the plants to stressful conditions. The article is well written both in English and in a proper scientific language, although in some cases scientific nomenclature is not correct and needs to be checked again to avoid low scientific sound.
Unfortunately, although the bibliographic references used to frame the problem are sufficient (but, more can be added), many of them are "old" and should be integrated with new ones. In many cases, the references indicated in the text are not present in the References list: authors are asked to check the correspondence of the references in the text and in the References list.
The title should be modified in its first part.
Figures should have better resolution and some captions need to be checked.
Figure S1 is not georeferenced. Please do this.
Check the term “gypsophilic value”: I think “gypsophily value” is better. Please consider changing "gypsophilic value" to "gypsophily value" throughout the whole manuscript.
Scientific names must be reported in italics in the text and in the other files.
Please, check and correct Table S2.
Please, in table S4 change "Species" to "Taxa", as subspecies are also listed there.
Please, change “locality” with “location” in Table 1, in the file "peerj-73212-data_paper_AndreuCera".
Please, the tables must be self-explanatory: therefore, the scientific names must be reported in full and completed with their authorship.

Experimental design

Lines 190-191: authors should explain how every species occurrence was recorded and species cover visually estimated.

Validity of the findings

The results are rich and well addressed in the Discussion. However, the Conclusions are too concise and need to be expanded upon.

Additional comments

Dear Authors

In general, I think your manuscript presents interesting results, discussed in enough detail. However, I think it doesn't sound very good from a scientific point of view and there are many aspects that you will need to review before it can be accepted for publication in a so good scientific journal. I took the liberty of making several corrections and adding some suggestions, thinking they will help you improve the text. Finally, I suggest that you broaden the Conclusions a little, in order to make the reader understand better what the message you want to send is.
Best regards.

Annotated reviews are not available for download in order to protect the identity of reviewers who chose to remain anonymous.

---

## Round 0.2 · accepted · Accept

Your revision has sufficiently addressed the Reviewers’ concerns. Your manuscript is now ready for publication.

Reviewer 1 ·

Basic reporting

The introductory section is well structured and justified with sufficient bibliographical references.

Experimental design

The objectives of the research work are well defined.

The methodology is clear and would be reproducible in any other study that matches the characteristics of this one.

Validity of the findings

The results obtained are significant and the conclusions have a two-way relationship with the objectives set.

Additional comments

I would like to thank the authors for taking all comments and suggestions into consideration. In my opinion the methodological aspects have been clarified with the changes made in this new version of the manuscript.

As I mentioned in the first revision of the draft, the general idea of the research work is very interesting, especially to observe how anthropic activities have been able to shape in a certain way some plant communities associated with a very arid environment, yet recognized as a Priority Habitat for conservation and with an extraordinary resilience capacity.